# Levofloxacin Versus Ciprofloxacin-Based Prophylaxis during the Pre-Engraftment Phase in Allogeneic Hematopoietic Stem Cell Transplant Pediatric Recipients: A Single-Center Retrospective Matched Analysis

**DOI:** 10.3390/antibiotics10121523

**Published:** 2021-12-14

**Authors:** Alessia G. Servidio, Roberto Simeone, Davide Zanon, Egidio Barbi, Natalia Maximova

**Affiliations:** 1Department of Medicine, Surgery and Health Sciences, University of Trieste, Piazzale Europa 1, 34127 Trieste, Italy; alessia.servidio@burlo.trieste.it (A.G.S.); egidio.barbi@burlo.trieste.it (E.B.); 2Department of Transfusion Medicine, ASUGI, Piazza dell’Ospitale 1, 34125 Trieste, Italy; roberto.simeone@asugi.sanita.fvg.it; 3Institute for Maternal and Child Health-IRCCS Burlo Garofolo, Via dell’Istria 65/1, 34137 Trieste, Italy; davide.zanon@burlo.trieste.it

**Keywords:** hematopoietic stem cell transplantation, pediatric, antibiotic prophylaxis, levofloxacin, ciprofloxacin

## Abstract

Infectious complications are the most common and significant cause of mortality and morbidity after allogeneic hematopoietic stem cell transplantation (HSCT). Antibacterial prophylaxis in pediatric cancer patients is a controversial issue. Our study compared the outcomes of levofloxacin versus ciprofloxacin prophylaxis in allogeneic HSCT pediatric recipients treated for hematological malignancies. A total of 120 patients received levofloxacin prophylaxis, and 60 patients received ciprofloxacin prophylaxis. Baseline characteristics such as age, gender, primary diagnosis, type of conditioning, donor type, stem cell source, and supportive care of the patients were similar, and duration of antibiotics prophylaxis was similar. Both prophylaxis regimens demonstrated the same efficacy on the risk of febrile neutropenia and severe complications such as sepsis, the same rate of overall mortality, hospital readmission, and length of hospital stay. Levofloxacin prophylaxis was associated with significantly lower cumulative antibiotic exposure. The median of Gram-positive infection-related antibiotic days was 10 days in the levofloxacin group versus 25 days in the ciprofloxacin group (*p* < 0.0001). The median of Gram-negative infection-related antibiotics was 10 days in the levofloxacin group compared with 20 days in the ciprofloxacin group (*p* < 0.0001). The number of days with body temperature ≥38 °C was significantly less in the levofloxacin group (*p* < 0.001).

## 1. Introduction

Despite the increasing success rate in allogeneic hematopoietic stem cell transplantation (HSCT), infections remain the most common and significant cause of mortality and morbidity after HSCT [1]. Bloodstream infections (BSI) are frequent and life-threatening complications in HSCT recipients, particularly during pre-engraftment [2,3,4]. The incidence of BSI in pediatric HSCT recipients is reported, in the literature, between 20% and 44% [5]. Approximately 40% of all BSI are related to at least one adverse outcome, constituting a significant burden to the pediatric HSCT population [6]. The BSI-associated mortality rate ranges from <5% in the case of Gram-positive bacteria to 40% in the case of multidrug-resistant (MDR) Pseudomonas aeruginosa and 64% in carbapenem-resistant Klebsiella pneumoniae infections [7,8,9]. Engraftment delay, myeloablative conditioning, severe mucosal damage, broad-spectrum antibiotics use, acute graft-versus-host disease (GVHD), prolonged corticosteroid treatments, and pre-transplant infectious history are the main risk factors for BSI and other severe infectious complications [10,11,12,13]. 

Due to their broad antimicrobial spectrum, fluoroquinolones have been a time-honored choice for prophylaxis in neutropenic patients with cancer. Additionally, studies have suggested that fluoroquinolones are more effective in reducing infections than placebo or no treatment, oral nonabsorbable antibiotics, or trimethoprim-sulfamethoxazole [14,15,16]. The European Conference on Infections approved antibacterial prophylaxis with fluoroquinolones in leukemia guidelines in 2007. The prophylaxis was recommended for high-risk neutropenic patients with an expected duration of neutropenia longer than seven days [17]. On the other side, the 8th European Conference on Infections in Leukemia (ECIL-8) group does not recommend routine antibacterial prophylaxis for pediatric with neutropenia during the pre-engraftment stage of HSCT, due to the risk of increased resistances [18].

Antibacterial prophylaxis in pediatric cancer patients is a controversial issue. As a result, most pediatric oncology societies do not provide recommendations on neutropenic individuals [6]. Only a few previous studies were conducted in pediatric patients with hematological cancer, but these did not provide conclusive results [19,20,21,22]. One recent randomized controlled trial evaluated the effect of levofloxacin prophylaxis in a large cohort of pediatric patients, demonstrating a significant BSI reduction in individuals with AML and relapsed ALL. No difference in BSI incidence was found in subjects undergoing HSCT [23].

However, to our knowledge, no studies compared the outcomes of levofloxacin versus ciprofloxacin prophylaxis in allogeneic HSCT pediatric recipients treated for hematological malignancies.

Our retrospective study aimed to evaluate potential differences in two prophylaxis regimens in this patient population.

## 2. Results

### 2.1. Patients and Clinical Characteristics

A total of 180 eligible pediatric HSCT recipients were enrolled in this study. A total of 120 patients received levofloxacin prophylaxis, and 60 patients received ciprofloxacin prophylaxis. Demographic characteristics of patients in both groups are described in Table 1. Within each group, baseline characteristics such as age, gender, primary diagnosis, type of conditioning, donor type, stem cell source, and supportive care of the patients were similar. We found no statistically significant difference in the median duration of neutropenia in both groups (*p* = 0.0779), while the median duration of aplasia was longer in patients receiving ciprofloxacin prophylaxis (*p* < 0.001). There was no difference in the median duration of levofloxacin prophylaxis compared with ciprofloxacin prophylaxis (12 days versus 10 days, *p* > 0.05). These results are shown in Table 1.

### 2.2. Treatment Outcomes

Patients undergoing ciprofloxacin prophylaxis had a similar risk of febrile neutropenia compared to patients receiving levofloxacin prophylaxis (33.3% versus 36.7%; *p* > 0.05). In contrast, no significant differences were found in the rate of severe sepsis, CLABSI, invasive fungal infection, and Gram-negative and polymicrobial bacteremia. Out of the 180 patients included in this study, 17 (23.3%) in the ciprofloxacin group and 18 (15%) in the levofloxacin group experienced at least one bacteremia event during the first 100 days after transplantation. In the first event analysis of available resistance data, we documented 7 (20%) fluoroquinolone-resistant strains out of a total of 35 events. We found no significant difference between the two groups: three strains (17.6%) were documented in the ciprofloxacin group and four strains (22.2%) in the levofloxacin group. The isolated fluoroquinolone-resistant organisms were E. coli, K. pneumoniae, E. cloacae, and P. aeruginosa. We documented a significantly lower rate of clinically documented infection in the levofloxacin group compared to the ciprofloxacin group (29.2% versus 53.3%; *p* < 0.05), as well as the incidence of Clostridium difficile-associated diarrhea (2.5% versus 9%; *p* < 0.05). In addition, the incidence of 30-days and 90-days in-hospital all-cause mortality and IRM was comparable in both groups. The primary and secondary outcomes are shown in Table 2. Patients in both groups had a similar length of hospital stay and a comparable all-cause and infection readmission rate within 90 days of transplantation (Table 1).

The Kaplan–Meier survival curve analysis revealed no statistically significant difference in in-hospital 90-day mortality between the two groups (Figure 1). Levofloxacin prophylaxis was associated with significantly lower cumulative antibiotic exposure. The median of Gram-positive infection-related antibiotic days was 10 days in the levofloxacin group versus 25 days (*p* < 0.0001) of the ciprofloxacin group (Figure 2a). Mean vancomycin exposure was 9 days in the levofloxacin group versus 23 days in the ciprofloxacin group (*p <* 0.001). A similar tendency was observed for days of Gram-negative infection-related antibiotics with a median of 10 days in the levofloxacin group compared with 20 days (*p* < 0.0001) of the ciprofloxacin group (Figure 2b). Finally, box-plot analysis in Figure 2c shows a statistically significant difference between the two groups for the number of days with body temperature ≥38 °C (*p* < 0.001).

Safety outcome specific to fluoroquinolones, such as musculoskeletal toxicity, were collected for this study, demonstrating the absence of side effects related to the use of both antibiotics.

## 3. Discussion

Infections represent one of the main complications of HSCT, not related to the patient’s primary disease, being associated with aplasia in the pre-engraftment phase. Between 30% and 50% of episodes of neutropenic fever have been microbiologically documented, and many of these are caused by bacteremia. Among the most frequent germs are Gram-positive bacteria, more often, coagulase-negative Staphylococcus and alpha-hemolytic Streptococci. However, Gram-negative bacteria represent the leading cause of severe sepsis, particularly Pseudomonas aeruginosa, related to mortality between 40% and 60% [24,25]. Therefore, prophylactic antibiotics were introduced in the pre-engraftment phase to prevent the onset of infections. To date, the most widely used antibiotics are fluoroquinolones. Their effectiveness was compared to other combinations, such as trimethoprim-sulphamethoxazole, oral nonabsorbable antibiotics, or placebo, highlighting lower rates of Gram-negative bacteremia, neutropenic fever, overall fever events, and non-relapse mortality. However, open and controlled studies showed a high rate of Gram-positive bacteremia among patients receiving prophylaxis with fluoroquinolones. Furthermore, some studies showed that the combination of fluoroquinolones with Gram-positive antibiotics reduces the incidence of Gram-positive bacteremia, mainly due to streptococcal species. Remarkably, the increasing incidence of fluoroquinolone-resistant strains correlates well with the growing use of these drugs in clinical practice. Immunocompromised patients, especially those with previous exposure to fluoroquinolones, will continue to become colonized and be at risk of developing bloodstream infections due to fluoroquinolone-resistant bacteria [26,27].

Our study analyzed the differences in outcomes between levofloxacin and ciprofloxacin prophylactic regimens in pediatric HSCT recipients. We found no statistically significant differences in the incidence of febrile neutropenia, the primary endpoint of our study. Three previous studies compared levofloxacin versus ciprofloxacin prophylaxis in HSCT recipients, in adults only, mainly in patients undergoing autologous transplantation, reporting controversial results. The first study described the same incidence of fever neutropenia in both groups [28]. In the second, the incidence was significantly fewer in the levofloxacin group [29], while the third study reported a lower incidence of febrile neutropenia in the ciprofloxacin group [30].

In our study, levofloxacin prophylaxis was associated with a nonsignificant difference in the duration of pre-engraftment neutropenia. Instead, the aplasia phase in this group was significantly shorter. As myelotoxicity is not a common side effect of either antibiotic, these findings are not likely to be related to the choice of the antibiotic prophylaxis regimen.

In contrast to the primary endpoint, we found a difference in the incidence of clinically documented infection and bloodstream infection in favor of levofloxacin prophylaxis, especially for those caused by Gram-positive organisms. The difference in the bloodstream infection rate was statistically significant, analyzing the first 30 days after transplantation, while this difference in the period before neutrophil engraftment was nonsignificant. This result could depend on the small sample size. We detected no differences in the number of Gram-negative bloodstream infections and infections complicated by sepsis in both groups. These results are similar to those reported by other authors [28,29]. 

Other studies revealed a decrease in the incidence of bacteremia in patients taking levofloxacin compared to other prophylaxis or placebo [31,32]. However, in the pediatric population, the literature presents no differences in the incidence of bacteremia between the group treated with levofloxacin and other prophylaxis, such as cefepime with or without vancomycin or no prophylaxis [5,20,22]. The study conducted by Alexander et al. proved a reduction in the incidence of bacteremia in children with acute myeloid leukemia receiving levofloxacin prophylaxis compared to the control group without prophylaxis (21.9% vs. 43.4%, *p* < 0.001). In contrast, no differences have been documented in autologous or allogeneic transplant recipients [23].

In our study, the levofloxacin prophylaxis group compared with the ciprofloxacin group demonstrated significant reductions in the use of Gram-negative empiric antibiotics, such as associations ceftazidime plus amikacin or meropenem plus amikacin. In addition, there was a significant trend toward a decrease in the use of vancomycin in the levofloxacin group. A potential explanation for these findings was that levofloxacin prophylaxis was associated with a lower incidence of the bloodstream and clinically documented infections, resulting in less use of first-line empiric antibiotics. We did not find studies comparing empiric antibiotic exposure between levofloxacin and ciprofloxacin prophylaxis. In an autologous HSCT setting, one study confronting antibiotic exposure between the levofloxacin prophylaxis group and non-prophylaxis groups reported the benefits of levofloxacin as well [33]. The incidence of acute GVHD was significantly lower in the levofloxacin group, with the data explained in the study published previously [34].

Finally, the finding that Clostridium difficile infection was less common among levofloxacin prophylaxis recipients was noted in other studies that compared levofloxacin with different prophylaxis regimens or non-prophylaxis protocols [22,29].

Our study has limits; first, we used a retrospective design, leading to more significant misclassification of the variables. Second, the presence of a small sample did not allow for accurate data analysis. In addition, the time frame considered in the study was quite broad. This could represent a bias since other changes in care related to infection prevention, and supportive treatments may have occurred, impacting the incidence of bacterial infections during the study period. However, since the primary outcome assessed was febrile neutropenia, only the impact of the two prophylaxis regimens was considered. We found no statistically significant difference for all secondary outcomes, which the new antibiotic therapies could have influenced.

Our study is the first to compare the effects of primary antibacterial prophylaxis with levofloxacin versus ciprofloxacin on serious infectious complications and antibiotic exposure in children undergoing allogeneic HSCT. Although both prophylaxis regimens demonstrated the same efficacy on the risk of febrile neutropenia and severe complications as sepsis, besides the same rate of overall mortality, hospital readmission, and length of stay, levofloxacin prophylaxis led to less exposure to antipseudomonal β-lactam antibiotics, aminoglycosides, vancomycin, and reduction of Clostridium difficile infection. Further study is required to compare the efficacy of levofloxacin versus ciprofloxacin prophylaxis on the largest group of pediatric allogeneic transplant recipients.

## 4. Materials and Methods

### 4.1. Study Design and Patients

A retrospective single-center study was conducted at the Pediatric Transplant Center of the Institute for Maternal and Child Health–IRCCS “Burlo Garofolo,” Trieste, Italy. The Institutional Review Board of the IRCCS Burlo Garofolo (reference no. RC 10/20) approved the study protocol. All parents of the patients gave written consent to collect and use personal data for research purposes.

We retrospectively reviewed medical records of all patients aged ≤18 years who underwent allogeneic HSCT between January 2005 and July 2020. Inclusion criteria were as follows: allogeneic HSCT for hematological malignancies, first transplant attempt, myeloablative conditioning regimen, and primary antibacterial prophylaxis during pre-engraftment phase with levofloxacin or ciprofloxacin. Inclusion criteria had no restriction on the cell source used, type of donor, and cellular graft composition. Patients who developed bacterial infection near the conditioning regimen and had undergone antibiotic treatment, patients who arrived at HSCT with aplasia or severe neutropenia, and patients who did not receive any antibiotic prophylaxis or received prophylaxis other than fluoroquinolones were excluded from the analysis.

As previously described, all transplant recipients were treated according to standard myeloablative conditioning protocols [35]. Standard GVHD prophylaxis included a calcineurin inhibitor to the sibling donor, associated with mycophenolate mofetil (MMF). Therapeutic drug monitoring-driven dosage was introduced in 2011 to the matched unrelated donor (MUD). MMF plus post-transplant cyclophosphamide was introduced in 2013 in the case of a haploidentical donor. All patients received prophylactic micafungin and acyclovir or valaciclovir during the peri-transplant period.

From 2005 to 2011, all transplant recipients underwent ciprofloxacin prophylaxis. From 2012 to nowadays, the institutional protocol switched to levofloxacin prophylaxis. Patients were treated with intravenous levofloxacin (10 mg/kg/dose twice daily, maximum 500 mg per dose) or intravenous ciprofloxacin (10 mg/kg/dose twice daily, maximum 500 mg per dose) from one day before their stem cell infusion until recovery from neutropenia. The dosage and method of administration of ciprofloxacin and levofloxacin remained unchanged over the years.

All patients underwent microbiological screening before transplantation. Patients colonized with fluoroquinolone-resistant germs were excluded from the study as they received physician-directed case-specific prophylaxis. Patients did not routinely receive fluoroquinolone prophylaxis for induction and consolidation chemotherapy before HSCT.

### 4.2. Febrile Neutropenia Treatment Protocol

At the first fever, defined as the axillary temperature of 38.0 °C or above, empiric antibiotic therapy, which included ceftazidime associated with amikacin, was initiated according to standardized pediatric national protocol discontinuing fluoroquinolone prophylaxis. Empiric antibiotic treatment was continued until neutrophil engraftment and then replaced with targeted antibiotic treatment for an identified infection.

### 4.3. Outcomes

We analyzed the outcomes by comparing the two periods: 2005 to 2011 (the ciprofloxacin group) versus 2012 to 2020 (the levofloxacin group). The primary outcome evaluated was febrile neutropenia, defined by a fever occurrence of ≥38.0 °C and an absolute neutrophil count <500 cells/μL. Secondary outcomes were assessed from day -1 through discharge and included rates of all febrile events, bloodstream, clinically or microbiologically documented infections, severe sepsis, central line-associated bloodstream infections (CLABSI), invasive fungal infections, Clostridium difficile infection rates, duration of antibiotic treatment against Gram-negative and Gram-positive bacteria, 30-day hospital readmission rate, 30-day and 90-day all-cause-mortality, and infection-related mortality (IRM). Length of stay was also evaluated as a secondary outcome and defined as hospital admission from the first day of conditioning until discharge. 

Aplasia was defined as an absence of almost all blood cellular components due to a lack of functioning stem cells during the pre-engraftment phase. Neutropenia was described as an absolute neutrophil count <500 cells/μL. A bloodstream infection event was designated as one positive blood culture obtained during a febrile episode, except for common skin commensals (coagulase-negative Staphylococci, Bacillus, and Corynebacterium spp.), which required at least two positive blood cultures to confirm bacteremia. Blood cultures (one aerobic and one anaerobic bottle per set) were obtained from each central venous catheter lumen. Subsequent blood cultures were drawn daily for persistent fever. 

Clinically documented infection was described as clinical or radiological signs that favored infection, but microbiological cultures were negative. Microbiologically documented infection was defined as bacterial, fungal, viral, or parasitic infection with supportive microbiological evidence, such as positive culture, antigen, or PCR test results, or characteristic histopathological findings [36]. Severe sepsis was referred to as infection in conjunction with severe dysfunction of cardiovascular or respiratory systems or ≥2 other organ systems [37]. The CLABSI definition was based on the Centers for Disease Control and Prevention/National Healthcare Safety Network (CDC/NHSN) guidelines [38]. Clostridium difficile infection was characterized as the identification of Cl. difficile toxin gene PCR in the presence of diarrhea. IRM was defined as the time from transplantation to death by an infectious cause, without relapse or recurrence and other transplant-related complications as competing events.

### 4.4. Statistical Analysis

Collected data were analyzed using descriptive statistics to determine the distribution and frequency of the variables. Continuous variables were expressed as median and confidence interval (CI) between second and third quartiles (percentile 25 and percentile 75), while categorical variables were expressed as the frequency and absolute or a percentage value. Box and whisker plots were generated for displaying the distribution of the numeric variable, and the Mann–Whitney test was used to compare the different groups of patients as appropriate. Two-tailed Fisher’s exact test was conducted to assess the association between categorical variables. Kaplan–Meier plots were generated for a graphical explanation of clinical outcomes to evaluate the differences in the infection overall survival between the two groups of antibacterial prophylaxis; the curves were compared with the log-rank test. *p*-values < 0.05 were considered as statistically significant. Statistical analyses were performed using WinStat (v.2012.1; In der Breite 30, 79189 Bad Krozingen, Germany) and MedCalc (Statistical Software version 18.9.1, Ostend, Belgium; http://www.medcalc.org; accessed on 1 September 2018).

## Figures and Tables

**Figure 1 antibiotics-10-01523-f001:**
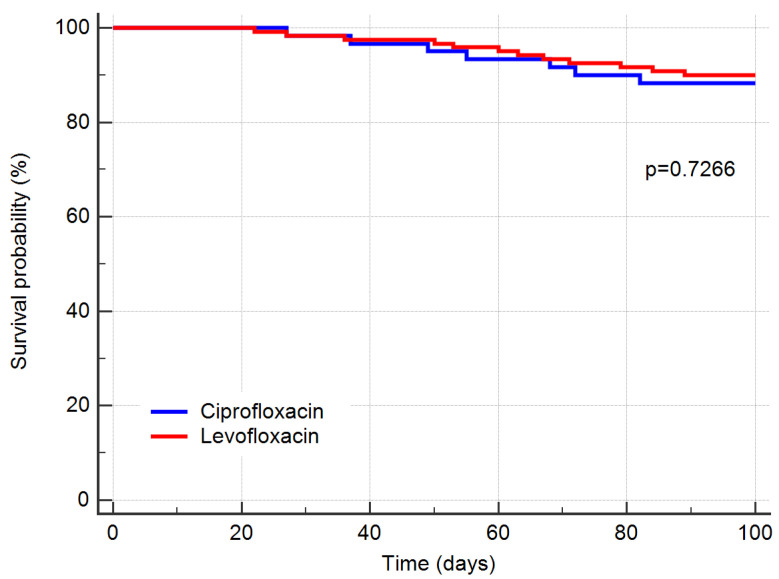
Kaplan–Meier survival curve analysis showed no statistically significant difference in in-hospital 90-day mortality between the levofloxacin prophylaxis and ciprofloxacin prophylaxis groups (*p* = 0.7266).

**Figure 2 antibiotics-10-01523-f002:**
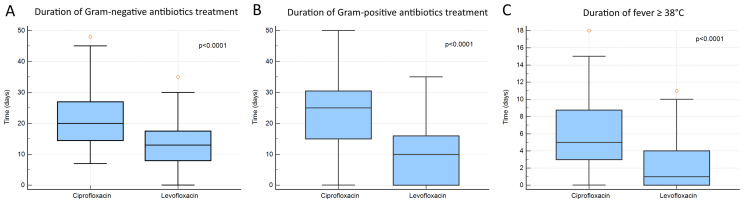
Box and whisker plot showing the statistically significant differences (*p* < 0.0001) in exposure to Gram-positive antibiotics between the levofloxacin and ciprofloxacin prophylaxis groups (**A**); box and whisker plot showing the statistically significant differences (*p* < 0.0001) in exposure to Gram-negative antibiotics between the levofloxacin and ciprofloxacin prophylaxis groups (**B**); box and whisker plot showing the statistically significant differences (*p* < 0.001) in an overall number of days with body temperature ≥38 °C between two groups (**C**).

**Table 1 antibiotics-10-01523-t001:** Patients baseline characteristics and transplant-related outcomes.

Baseline Characteristics	Ciprofloxacin Group	Levofloxacin Group	*p*-Value
Number of patients (%)	60 (33.3)	120 (66.7)	-
Gender, male/female, number (%)	39/21 (65/35)	76/44 (64.3/36.7)	0.8704
Age, median (IQR), years	8.5 (4–13)	8.0 (4–13)	0.9854
Primary diagnosis, number (%):			
acute lymphoblastic leukemia	30 (50.0)	57 (47.5)	0.7548
acute myeloid leukemia	11 (18.3)	14 (11.7)	0.2555
myelodysplastic syndrome	1 (1.7)	5 (4.2)	0.6625
solid tumor	13 (21.7)	28 (23.3)	0.8525
non-malignant disease	5 (8.3)	16 (13.3)	0.4608
Allogeneic transplant, number (%)	60 (100)	120 (100)	-
Myeloablative conditioning, number (%):	60 (100)	120 (100)	-
chemotherapy-based	36 (60.0)	67 (55.8)	0.6341
TBI-based	24 (40.0)	53 (44.2)	0.6344
ATG use, number (%)	39 (65.0)	81 (67.5)	0.7403
Graft cell dose, median (IQR)			
CD34 + cells × 10^6^/kg	8.6 (5.7–11.1)	7.5 (5.9–10.5)	0.3973
TNC × 10^8^/kg	5.4 (4.5–8.2)	5.6 (4.8–8.1)	0.8156
Duration of neutropenia, median (IQR), days	18 (15–20)	16 (13–19.7)	0.0779
Duration of aplasia, median (IQR), days	11 (10–12)	10 (9–11)	<0.001
Duration of prophylaxis, median (IQR), days	10 (7.2–14)	12 (8–17)	0.3475
Supportive care interventions, number (%):			
prophylactic G-CSF	16 (26.7)	28 (23.3)	0.7133
steroids for >10 days consecutively	25 (41.7)	47 (39.2)	0.7498
steroids ≥ 2 mg/kg >7 days consecutively	16 (26.7)	29 (24.2)	0.7184
Acute GVHD grade II-IV, number (%):	32 (53.3)	9 (7.5)	<0.0001
Length of stay, median (IQR), days	43.5 (38–48)	42 (37–48)	0.6236
Readmission,* number (%)	22 (36.6)	37 (30.8)	0.501
Infection-related readmission,* number (%)	9 (15.0)	11 (9.2)	0.314

* Readmission within 90 days of transplantation. IQR, interquartile range; ATG, anti-thymocyte globulin; TNC, total nuclear cells; G-CSF, granulocyte colony-stimulating factor; GVHD, graft-versus-host disease.

**Table 2 antibiotics-10-01523-t002:** Comparison of infection-related complications in the ciprofloxacin and the levofloxacin prophylaxis groups.

Outcomes	Ciprofloxacin Group (*n* = 60)	Levofloxacin Group (*n* = 120)	*p*-Value
Febrile neutropenia, number (%)	22 (36.7)	40 (33.3)	0.7397
Bloodstream infection, number (%):	17 (28.3)	18 (15.0)	<0.05
at the first episode of febrile neutropenia	9 (15.0)	6 (5.0)	<0.05
within 30 days of transplantation	15 (25.0)	14 (11.7)	<0.05
before neutrophil engraftment	12 (20.0)	11 (9.2)	0.0567
associated with severe sepsis	5 (8.3)	4 (3.3)	0.1624
Gram-positive bacteremia	12 (20.0)	10 (8.3)	<0.05
Gram-negative bacteremia	7 (11.7)	9 (7.5)	0.408
Polymicrobial	2 (3.3)	2 (1.7)	0.6016
CLABSI	5 (8.3)	6 (5.0)	0.51
Clinically documented infection, number (%)	32 (53.3)	35 (29.2)	<0.05
Invasive fungal infection, number (%)	8 (13.3)	9 (7.5)	0.783
Clostridium difficile infection, number (%)	9 (15.0)	3 (2.5)	<0.05
Overall antibiotic exposure, median (IQR), days:			
within day + 30	21 (16–25)	13 (9–19)	<0.0001
within day + 100	38 (34.5–41.5)	31 (31–33)	<0.05
90-day overall mortality, number (%):	6 (1.0)	10 (8.3)	1
infection-related	2 (3.3)	4 (3.3)	1
bacteria-related	1 (1.7)	2 (1.7)	1
30-day overall mortality, number (%):	1 (1.7)	2 (1.7)	1
infection-related	1 (1.7)	1 (0.8)	1
bacteria-related	1 (1.7)	1 (0.8)	1

CLABSI, central line-associated bloodstream infections; IQR, interquartile range.

## Data Availability

The data presented in this study are available on request from the corresponding author. The data are not publicly available due to the impracticality of public access.

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
