# Peer review of "Levofloxacin Versus Ciprofloxacin-Based Prophylaxis during the Pre-Engraftment Phase in Allogeneic Hematopoietic Stem Cell Transplant Pediatric Recipients: A Single-Center Retrospective Matched Analysis"

_antibiotics, 2021, doi:10.3390/antibiotics10121523_

Round 1
Reviewer 1 Report
This is an original research article addressing the impact of levofloxacin-based prophylaxis during the pre-engrafment phase in pediatric patients undergoing hematopoietic stem cell transplantation. The study retrospectively analyzes, as a primary endpoint, the difference in the incidence of febrile neutropenia in patients receiving ciprofloxacin vs levofloxacin. Although no differences were found for the primary outcome, Authors found that levofloxacin prophylaxis was associated with significantly lower cumulative antibiotic exposure and a lower median Gram-positive infection-related antibiotic days.
The authors declare that "This result could depend on the small sample size.", maybe a possible explanation in the reduction of GRAM + BSI could hypothesized.
Thus, what this manuscript add to our knowledge?
The topic is interesting since final evidence on the use of fluoroquinolones in a prophylactic setting in pediatric HSCT is missing; however, I have some concerns on the study design and about the reliability of the data, also considering the recent data of Alexander et al. on the role of Ab prophylaxis in children receiving HSCT. I would recommend major revisions to the paper.
Major concerns
- Among the 180 patients enrolled in the study, 120 received levofloxacin and 60 received ciprofloxacin. I would recommend specifying what the rationale for the use of the one or the other was. This could represent a very important bias. Patients were randomly assigned? Moreover, it would be very important to report whether the administration of levofloxacin or ciprofloxacin changed in the 15 years of the study. Were the patients screened before transplant for fluoroquinolones-resistant strain (or resistant to other antibiotics)? If so, I would suggest adding this information in the text.
- The timespan of enrolment represents a major concern. Outcomes of patients receiving HSCT from 2005 to 2020 quite changed and thus comparison of outcomes between the two groups seems to lose significance. A sub analysis for different time-periods should be presented (e.g., at least two periods 2005-2012 vs 2012-2020).
- Total days of antibiotic exposure within day +30 and day +100 of the two groups should be included to assess any confounding factor.
- Considering that one of the main drawbacks of quinolones is the antibiotic resistance, it would be of interest to add the resistance pattern of isolated pathogens.
- May the authors better specify in the methods how do they define aplasia in contrast to neutropenia?
- The data on aGvhD should be added in table 1
Minor concerns
- The title does not quite reflect the content of the paper. I would suggest modifying it.
- There are some minor language errors in the manuscript (e.g., line 53 “engraftment delayed”, line 58 “long for a long time”). I would recommend fixing them.
- In the introduction I would suggest also discussing the 8th ECIL guidelines reporting the current indications for antibiotic prophylaxis in pediatric patients undergoing HSCT.
- In Table 1 and Table 2 data in the first and the second column are not perfectly alienated. I would recommend adjusting it. I would also add in Table 2 the data regarding incidence of bacteremia in the first episode of febrile neutropenia.
- In Table 1 I would reason whether the duration of neutropenia, duration of aplasia, length of stay, re-admission and infection related re-admission represent baseline characteristics or rather results. Moreover, considering that days of aplasia represented a study end-point I would add the number of infused stem cells and, if applicable, the use of ATG.
- Both line 101 and 104 starts with “In addition”, I would recommend avoiding repetitions.
- I would suggest expanding the discussion section regarding lines 143-147
- Line 174 I would recommend adding the description of the treatment received by the control group of the cited study (no antibiotic?).
- Line 179 the authors discussed the reduction in vancomycin administration in the levofloxacin group, but I could not find this data in the results section. I would suggest mentioning it.
- Were there some differences in the duration of febrile neutropenia? Considering that febrile neutropenia is quite almost present in pediatric patients receiving myeloablative regimens it sounds quite odd to me that only 36% and 33% of patients respectively presented it.
- References 31 and 33 are the same.
Author Response
Major concerns
- Among the 180 patients enrolled in the study, 120 received levofloxacin and 60 received ciprofloxacin. I would recommend specifying what the rationale for the use of the one or the other was. This could represent a very important bias. Patients were randomly assigned? Moreover, it would be very important to report whether the administration of levofloxacin or ciprofloxacin changed in the 15 years of the study. Were the patients screened before transplant for fluoroquinolones-resistant strain (or resistant to other antibiotics)? If so, I would suggest adding this information in the text.
We agree with the reviewer's comment, and we have modified the paragraph detailing antibiotic prophylaxis in the Materials and Methods chapter (lines 248 - 256).
From 2005 to 2011, all transplant recipients underwent ciprofloxacin prophylaxis. From 2012 to nowadays, the institutional protocol switched to levofloxacin prophylaxis. The dosage and method of administration of ciprofloxacin and levofloxacin remained unchanged over the years.
All patients underwent microbiological screening before transplantation. Patients colonized with fluoroquinolone-resistant germs were excluded from the study as they received physician-directed case-specific prophylaxis.
The timespan of enrolment represents a major concern. Outcomes of patients receiving HSCT from 2005 to 2020 quite changed and thus comparison of outcomes between the two groups seems to lose significance. A sub analysis for different time-periods should be presented (e.g., at least two periods 2005-2012 vs 2012-2020).
We agree with the reviewer’s comment. Indeed, our analysis compares the results between the ciprofloxacin and levofloxacin groups, thereby comparing the transplant outcomes over two time periods (2005-2011 versus 2012-2020). We have specified this in the Materials and Methods (lines 265 - 266).
- Total days of antibiotic exposure within day +30 and day +100 of the two groups should be included to assess any confounding factor.
We added the overall antibiotic exposure in two groups within day +30 and day +100 in Table 2.
- Considering that one of the main drawbacks of quinolones is the antibiotic resistance, it would be of interest to add the resistance pattern of isolated pathogens.
We added the data about fluoroquinolone-resistant organisms in the Results section (lines 107 - 113).
Of the 180 patients included in this study, 17 (23.3%) patients in the ciprofloxacin group and 18 (15%) in the levofloxacin group experienced at least one bacteremia event during the first 100 days after transplantation. In the first event analysis of available resistance data, we documented 7 (20%) fluoroquinolone-resistant strains out of a total of 35 events. We found no significant difference between the two groups: 3 strains (17.6%) were documented in the ciprofloxacin group and four strains (22.2%) in the levofloxacin group. The isolated fluoroquinolone-resistant organisms were E. coli, K. pneumoniae, E. cloacae, and P. aeruginosa.
- May the authors better specify in the methods how do they define aplasia in contrast to neutropenia?
We added the required definitions in the Materials and Methods section (lines 275 - 277).
Aplasia was defined as an absence of almost all blood cellular components due to a lack of functioning stem cells during the pre-engraftment phase. Neutropenia was defined as an absolute neutrophil count < 500 cells/μL.
- The data on aGvhD should be added in table 1
We added the data of a GVHD rate in both groups in Table 1. We explained the differences in the incidence of a GVHD in a previously published manuscript (Carlone G, Simeone R, Baraldo M, Maestro A, Zanon D, Barbi E, Maximova N. Area-under-the-Curve-Based Mycophenolate Mofetil Dosage May Contribute to Decrease the Incidence of Graft-versus-Host Disease after Allogeneic Hematopoietic Cell Transplantation in Pediatric Patients. J Clin Med. 2021 Jan 21;10(3):406).
We have added this explanation in the Material and Method and the Discussion section (lines 242 – 245 and 205 - 206).
Minor concerns
- The title does not quite reflect the content of the paper. I would suggest modifying it.
We modified the title to better reflect the paper contents: “Levofloxacin versus ciprofloxacin-based prophylaxis during the pre-engraftment phase in allogeneic hematopoietic stem cell transplant pediatric recipients: a single-center retrospective matched analysis.”
- There are some minor language errors in the manuscript (e.g., line 53 “engraftment delayed”, line 58 “long for a long time”). I would recommend fixing them.
We corrected these mistakes.
- In the introduction I would suggest also discussing the 8thECIL guidelines reporting the current indications for antibiotic prophylaxis in pediatric patients undergoing HSCT.
We added the sentence reporting the 8th ECIL guidelines in the Introduction (lines 64 - 67).
- In Table 1 and Table 2 data in the first and the second column are not perfectly alienated. I would recommend adjusting it. I would also add in Table 2 the data regarding the incidence of bacteremia in the first episode of febrile neutropenia.
We adjusted both tables and added the data regarding the incidence of bacteremia during the first episode of febrile neutropenia.
- In Table 1 I would reason whether the duration of neutropenia, duration of aplasia, length of stay, re-admission and infection related re-admission represent baseline characteristics or rather results. Moreover, considering that days of aplasia represented a study end-point I would add the number of infused stem cells and, if applicable, the use of ATG.
We agree with the reviewer’s comment. We changed the caption of Table 1. Moreover, we added the data of graft composition and use of ATG.
- Both line 101 and 104 starts with “In addition”, I would recommend avoiding repetitions.
We deleted “in addition” in the first sentence.
- I would suggest expanding the discussion section regarding lines 143-147
We added an explanation, as suggested, in the Discussion section (lines 157 - 167).
- Line 174 I would recommend adding the description of the treatment received by the control group of the cited study (no antibiotic?).
We added no prophylaxis received by the control group (lines 192 - 194).
- Line 179 the authors discussed the reduction in vancomycin administration in the levofloxacin group, but I could not find this data in the results section. I would suggest mentioning it.
We added the data about vancomycin exposure in the Result section (lines 129 - 130).
- Were there some differences in the duration of febrile neutropenia? Considering that febrile neutropenia is quite almost present in pediatric patients receiving myeloablative regimens it sounds quite odd to me that only 36% and 33% of patients respectively presented it.
We understand the reviewer's confusion, but these are numbers from our case series. We cannot change them. Anyway, we have redone the count. The numbers are correct.
- References 31 and 33 are the same.
We deleted reference 33.
Reviewer 2 Report
Comments to the Author:
This manuscript is a retrospective single center report of 180 pediatric HSCT recipients treated with levofloxacin (120) or ciprofloxacin (80) prophylactic regimens. Previous limited studies in adults reported controversial results. With the limit of a retrospective design, this study compare the difference in outcomes between levofloxacin and ciprofloxacin prophylaxis, showing no statistically difference in the incidence of febrile neutropenia (primary endpoint) but a difference in incidence of clinically documented infection and bloodstream infection in favor of levofloxacin prophylaxis.
Major revision:
- Materials and methods: Considering that routine levofloxacin was introduced in 2014, please specify if patients who received ciprofloxacin were treated previously or if the two arms were equally distributed in the period “2005-2020”. If patients receiving ciprofloxacin were mostly in a previous period, differences could be related to new treatment antibiotics? Please argument also in “treatment outcomes”
Minor revions:
- Line 5 of the introduction: consider adding pediatric references (i.e. Dandoy CE, Pediatr Blood Cancer. 2019 Dec;66(12):e27978)
- Line 13 of the introduction: consider adding more recent references (i.e.Efficacy of levofloxacin as an antibacterial prophylaxis for acute leukemia patients receiving intensive chemotherapy: a systematic review and meta-analysis; Weerapat Owattanapanich Meta-Analysis Hematology 2019 Dec;24(1):362-368. Comparative Study Cancer Med; Efficacy of antibiotic prophylaxis in patients with cancer and hematopoietic stem cell transplantation recipients: A systematic review of randomized trials. Grace Egan. 2019 Aug;8(10):4536-4546)
- Please specify if ciprofloxacin was administered intravenous
- Please specify if evaluation of antibiotic resistance was differently observed in the levofloxacin and ciprofloxacin arms
Author Response
Major revision:
- Materials and methods: Considering that routine levofloxacin was introduced in 2014, please specify if patients who received ciprofloxacin were treated previously or if the two arms were equally distributed in the period “2005-2020”. If patients receiving ciprofloxacin were mostly in a previous period, differences could be related to new treatment antibiotics? Please argument also in “treatment outcomes”
Levofloxacin has been used in our Institution since 2000 (Levoxacin, GlaxoSmithKline S.p.A).
We have modified the paragraph explaining antibiotic prophylaxis in the Materials and Methods chapter (lines 248 - 256).
From 2005 to 2011, all transplant recipients underwent ciprofloxacin prophylaxis. Since 2012 nowadays, the institutional protocol has provided for levofloxacin prophylaxis. The dosage and method of administration of ciprofloxacin and levofloxacin remained unchanged over the years.
All patients underwent microbiological screening before transplantation. Patients colonized with fluoroquinolone-resistant germs were excluded from the study as they received physician-directed case-specific prophylaxis. Indeed, our analysis compares the results between the ciprofloxacin and levofloxacin groups, thereby comparing the transplant outcomes over two time periods (2005-2011 versus 2012-2020).
Since the primary outcome was febrile neutropenia, only the impact of the two prophylaxis regimens was considered. We found no statistically significant difference for all secondary outcomes, which the new antibiotic therapies could influence. We explained this argument in the Discussion section (lines 212 - 217).
Minor revions:
- Line 5 of the introduction: consider adding pediatric references (i.e. Dandoy CE, Pediatr Blood Cancer. 2019 Dec;66(12):e27978)
- Line 13 of the introduction: consider adding more recent references (i.e.Efficacy of levofloxacin as an antibacterial prophylaxis for acute leukemia patients receiving intensive chemotherapy: a systematic review and meta-analysis; Weerapat Owattanapanich Meta-Analysis Hematology 2019 Dec;24(1):362-368. Comparative Study Cancer Med; Efficacy of antibiotic prophylaxis in patients with cancer and hematopoietic stem cell transplantation recipients: A systematic review of randomized trials. Grace Egan. 2019 Aug;8(10):4536-4546)
We added all references requested.
- Please specify if ciprofloxacin was administered intravenous
We have specified the method of administration in the Materials section (lines 250 – 251).
- Please specify if evaluation of antibiotic resistance was differently observed in the levofloxacin and ciprofloxacin arms
We added the data about fluoroquinolone-resistant organisms in the Results section (lines 107 - 113).
Of the 180 patients included in this study, 17 (23.3%) patients in the ciprofloxacin group and 18 (15%) in the levofloxacin group experienced at least one bacteremia event during the first 100 days after transplantation. In the first event analysis of available resistance data, we documented only 7 (20%) fluoroquinolone-resistant strains of a total of 35 events. We found no significant difference between the two groups: 3 strains (17.6%) were documented in the ciprofloxacin group and four strains (22.2%) in the levofloxacin group. The isolated fluoroquinolone-resistant organisms were E. coli, K. pneumoniae, E. cloacae, and P. aeruginosa.
Round 2
Reviewer 1 Report
No others comment
Reviewer 2 Report
This manuscript is a retrospective single center report of 180 pediatric HSCT recipients treated with levofloxacin (120) or ciprofloxacin (80) prophylactic regimens. Previous limited studies in adults reported controversial results. With the limit of a retrospective design, this study compare the difference in outcomes between levofloxacin and ciprofloxacin prophylaxis, showing no statistically difference in the incidence of febrile neutropenia (primary endpoint) but a difference in incidence of clinically documented infection and bloodstream infection in favor of levofloxacin prophylaxis. Authors have satisfactorily implemented requested revisions in the paper.I have no other revisions to suggest.